# Nutritional Status Assessed with Objective Data Assessment Correlates with a High-Risk Foot in Patients with Type 2 Diabetes

**DOI:** 10.3390/jcm11051314

**Published:** 2022-02-27

**Authors:** Yusuke Mineoka, Michiyo Ishii, Yoshitaka Hashimoto, Aki Yamashita, Takahiro Takemura, Sachiyo Yamaguchi, Machiko Toyoda, Michiaki Fukui

**Affiliations:** 1Department of Internal Medicine, Otsu City Hospital, Otsu 520-0804, Japan; mineoka8@koto.kpu-m.ac.jp (Y.M.); michiishii@gmail.com (M.I.); take_take_take_taka@yahoo.co.jp (T.T.); victoria2374@outlook.jp (S.Y.); toyoda26@koto.kpu-m.ac.jp (M.T.); 2Department of Endocrinology and Metabolism, Graduate School of Medical Science, Kyoto Prefectural University of Medicine, Kyoto 602-8566, Japan; y-hashi@koto.kpu-m.ac.jp; 3Department of Nursing, Otsu City Hospital, Otsu 520-0804, Japan; chacoraki@gmail.com

**Keywords:** clinical practice, diabetes, foot risk, nutritional status

## Abstract

Malnutrition and diabetes are likely to co-occur. There are few reports on the association between nutritional status and foot risk in patients with type 2 diabetes (T2D). Therefore, we aimed to investigate this relationship in this cross-sectional study. We investigated the relationships between objective data assessment (ODA), especially Controlling Nutritional Status (CONUT) score and foot risk, evaluated by the International Working Group on the Diabetic Foot (IWGDF), in consecutive patients with T2D. Patients were divided into groups 0 to 3 by IWGDF, and groups 1 to 3 were defined as high-risk groups. Among 469 patients, 42.6% (*n* = 200) of them had high-risk foot. Patients with high-risk foot were significantly older (71.2 ± 11.3 vs. 64.2 ± 13.4 years, *p* < 0.001) and had a longer duration of diabetes (18.0 ± 12.0 vs. 11.5 ± 10.0 years, *p* < 0.001) than those in the low-risk group. In the high-risk group, serum albumin level, total lymphocyte count, hemoglobin, and CONUT score were significantly worse, especially in older patients (≥75 years). Multivariate logistic regression analysis showed that there was a positive correlation between CONUT score and high-risk foot in older patients (OR, 1.37; 95% CI, 1.05–1.86; *p* = 0.021). Our results indicated that nutritional status, assessed by ODA, correlated with high-risk foot, especially in older patients with T2D.

## 1. Introduction

Diabetic foot ulcers and gangrene are known to be caused by diabetic peripheral neuropathy (DPN) and peripheral artery disease (PAD), which affects 25% of patients with diabetes [1,2]. The infection causes or worsens foot ulcers due to complications of DPN and PAD [3]. Moreover, amputation of lower limbs due to diabetic foot ulcer and gangrene reduces patients’ quality of life and results in a physical and mental burden on them and their families, as well as a huge financial burden on society [4]. Therefore, the importance of foot screening and foot care in patients with diabetes is widely recognized.

Malnutrition is influenced by several factors, and the nutritional status of patients with diabetes worsens due to diabetic complications and comorbidities [5]. Malnutrition worsens underlying diseases and leads to unfavorable prognosis in older patients with diabetes [6]. Malnourished patients with diabetes have been shown to be twice as likely to have foot injuries compared with nourished patients [6]. Maintaining and improving nutritional status is important in the treatment of foot ulcers and gangrene [7]; however, there are few reports on the relationship between nutritional status and the risk of diabetic foot in patients with diabetes. Therefore, we performed a cross-sectional study of patients with type 2 diabetes to investigate the relationship between nutritional status, assessed using an objective data assessment (ODA), and diabetic foot risk, proposed by the International Working Group on the Diabetic Foot (IWGDF) [8].

## 2. Materials and Methods

### 2.1. Study Participants and Data Collection

We performed this study in accordance with the Declaration of Helsinki and obtained informed consent from all patients. This study was approved by the Ethics Committee of Otsu City Hospital (No. 213). We included patients with type 2 diabetes >20 years of age who were the outpatients of Otsu City Hospital (Otsu, Japan) and whose legs were examined and tested. Patients were assessed for smoking status using a self-administered questionnaire.

Blood samples were gathered in the morning after an overnight fast to measure hemoglobin (g/dL), total lymphocyte count (count/mL), hemoglobin A1c (%), creatinine (µmol/L), total cholesterol (mmol/L), cholinesterase (U/L), and serum albumin (g/dL). Complete blood counts and examinations were performed using a Beckman Coulter LH 780 instrument and Bio Majesty JCA-BM 6050 (JEOL, Tokyo, Japan). The Controlling Nutritional Status (CONUT) score was calculated using the data of serum albumin levels, total cholesterol levels, and total lymphocyte counts [9]; albumin levels ≥3.5, <3.5 and >3.0, <2.99 and ≥2.5, and <2.5 g/dL were scored as 0, 2, 4, and 6 points, respectively; total lymphocyte count of ≥1600, 1599–1200, 1199–800, and <800/mm^3^ were scored as 0, 1, 2, and 3 points, respectively; and total cholesterol levels ≥180, 140–179, 100–139, and <100 mg/dL were scored as 0, 1, 2, and 3 points, respectively. The CONUT score was defined as the sum of scores, ranging from 0 to 12, with higher scores indicating a worse nutritional status.

Patients with acute inflammatory or infectious diseases, hematological diseases, malignancy, severe organ damage, including nephrotic syndrome or liver cirrhosis, or blood diseases were excluded from our study.

Type 2 diabetes was diagnosed as previously reported [10], and diabetic foot risk was categorized into groups using the IWGDF classification, as follows [8]: (0) (no loss of protective sensation (LOPS) and no peripheral artery disease (PAD)), (1) (LOPS or PAD), (2) (LOPS + PAD or LOPS + foot deformity or PAD + foot deformity), and (3) (LOPS or PAD and one or more of the following: history of a foot ulcer, a lower-extremity amputation (minor or major) and end-stage renal disease). We defined groups 1–3 as the high-risk group according to a previous report [11]. Examination of the lower limbs was performed by a certified nurse for diabetes nursing, a diabetologist, or a certified diabetes educator. DPN was diagnosed using the diagnostic criteria for diabetic neuropathy proposed by the Diagnostic Neuropathy Study Group [12]. Two or more abnormalities of three examination items were used to diagnose DPN: neuropathic symptoms such as neuropathic pain, paresthesia and numbness, decreased or absent ankle reflex (bilateral), and decreased distal sensation assessed by C128 Hz tuning fork without evident non-diabetic peripheral neuropathy. Diabetic retinopathy was diagnosed by an ophthalmologist as previously reported [13], and diabetic nephropathy was defined as nephropathy with urine microalbuminuria >30 mg/gCre [14]. PAD was diagnosed if at least one of the following was confirmed: ankle brachial pressure index (ABI) < 0.9 or absence of two or more pedal pulses on palpation. Foot deformity and musculoskeletal abnormalities were examined to detect hallux valgus deformity, hammer/claw toe deformity, and hallux limitus (limited motion at the metatarsophalangeal joint). Stratified analysis was performed between older (≥75 years) and younger patients. We divided the patients according to statin use because statin usage decreases total cholesterol levels, which leads to increased CONUT scores. 

### 2.2. Statistical Analysis

For statistical analysis, JMP v.9.0 (SAS Institute Inc., Cary, NC, USA) was used, and statistical significance was set at *p* < 0.05. A chi-square test, unpaired Student’s *t*-test or analysis of variance, or post hoc Tukey–Kramer test was used for comparison analyses between the groups. The data were analyzed by cross-tabulation, Pearson χ2 test, or Fisher’s exact test. Multivariate logistic regression analysis was used to adjust for factors associated with nutritional status and high-risk foot. We selected covariates for multivariate analysis, including sex, BMI, age, duration of diabetes, current smoking status, creatinine level, HbA1c level, and hypertension.

## 3. Results

In this study, a total of 553 patients were included. Among them, 68 patients were excluded because of malignancy or blood diseases (*n* = 27), foot ulcers (*n* = 17), severe tissue damage (*n* = 12), liver cirrhosis (*n* = 4), acute inflammatory or infectious disease (*n* = 3), nephrotic syndrome (*n* = 3), and acute massive hemorrhage (*n* = 2). The clinical characteristics of study participants according to the IWGDF criteria are described in Table 1. Patients in group 1 and group 2 assessed using the IWGDF criteria were significantly older and had a longer duration of diabetes than those in group 0. Total cholesterol was significantly worse in group 2, and cholinesterase was significantly worse in group 1. Serum albumin level, hemoglobin, and CONUT scores were significantly worse in group 1 and 2.

In Table 2, the clinical characteristics of study participants were compared according to low- or high-risk IWGDF criteria. Patients with high-risk foot, assessed using the IWGDF criteria, were significantly older and had a longer duration of diabetes than those in the low-risk group. Total lymphocyte count, hemoglobin, cholinesterase, serum albumin level, and CONUT scores were significantly worse in the high-risk foot group (Table 2).

A comparative analysis indicated that older patients (≥75 years) had worse nutritional status, as assessed by several ODAs, whereas no significant difference was found in glycemic status and the proportion of statin use. The proportions of hypertension and microangiopathy were higher in older patients (Table 3).

A stratified analysis between older (≥75 years) and younger groups showed that serum albumin was significantly low in group 3 and hemoglobin was significantly low in group 1 in the older group (Table 4a). In the group younger than 75 years of age, serum albumin levels in group 1 and 2 were low and hemoglobin was low in group 1 at significant levels (Table 4b). In all age groups, there were no significant differences in BMI and HbA1c with or without foot risk. The proportions of hypertension and nephropathy had significant differences in each group, and the disease duration was significantly longer in group 2 in both the older and younger groups (Table 4a,b).

Multivariate logistic regression analyses showed that the CONUT score was associated with a high-risk foot in the older group, after adjusting for several factors. This relationship was not observed in the younger group of patients (Table 5). Moreover, multivariate logistic regression analyses showed a correlation between CONUT score and high-risk foot in the older group, regardless of statin use (Appendix A). This relationship was not observed in the younger group (Appendix A).

## 4. Discussion

This study revealed that patients with type 2 diabetes and with high-risk foot were older, had a longer duration of diabetes, had poor glycemic control, and had a worse renal function. In addition, their nutritional status, as assessed by ODAs, was significantly worse, especially in older patients.

Diabetes is often associated with malnutrition, especially in older patients, and the association has been previously reported [15,16,17,18]. Malnutrition in patients with diabetes and high-risk foot is known to be associated with inflammation-related atherosclerosis, leading to amputation of the lower extremities in addition to known risk factors [19]. Therefore, timely nutritional assessment is needed for patients with diabetes and high-risk foot. Although Mini Nutritional Assessment (MNA) and subjective global assessment (SGA) are well-known nutritional assessment screening tools [20,21], they are not always easy to perform routinely in clinical practice. SGA is a well-established tool for nutritional assessment [20]; however, it is subjective and requires an evaluator with some training and specialized knowledge for accurate assessment. MNA is an excellent nutritional assessment tool for older individuals [21], but it is relatively time-consuming because of many questions.

On the other hand, ODA is useful for nutritional evaluation in daily medical care because it is relatively easy to obtain and cost-effective. Serum albumin level and BMI are well-known markers of malnutrition, and the relationship between malnutrition and total mortality has been reported in older people [22,23]. Moreover, a serum albumin level of <3.5 g/dl has been shown to correlate with decreased visceral protein [24] and is reported to be an independent risk factor of all-cause mortality [25]. However, physicians should be cautious in evaluating nutritional status with serum albumin levels because of the effect of age and various conditions, including inflammation and liver or kidney diseases [26,27]. BMI is an important index in patients with diabetes; however, a previous report indicated that >30% of patients with diabetes diagnosed with malnutrition had a BMI ≥ 30 kg/m^2^ [28]. Therefore, it may be difficult to evaluate the nutritional status of patients with diabetes by BMI alone. In this study, BMI was lower in older patients; however, no significant differences were found between the high-risk and low-risk foot groups at all ages.

CONUT is a complex ODA, calculated using total lymphocyte count, total cholesterol level, and serum albumin level [9]. CONUT evaluates nutritional status from various perspectives using three types of objective biomarkers: protein metabolism, immune function, and lipid metabolism [9]. A positive relationship between CONUT score and SGA was also reported previously [29]. In addition, previous studies showed that the CONUT score is a useful marker for mortality [30,31], healing of foot ulcers [32,33], and subclinical atherosclerosis [34]. In the present study, multivariate logistic regression analysis indicated that the CONUT score was significantly associated with a high-risk foot in the older group, with or without statin use. Serum albumin levels were low in the high-risk foot group in all age groups, but CONUT was significantly poor in the high-risk foot group only in the older group in this study. Since the limitation of nutritional assessment with serum albumin level alone has been pointed out [26,27], it might indicate severe malnutrition in the high-risk foot group in older patients.

All ODAs were poor, and the microvascular complications of diabetes were advanced in the older group with the high-risk foot; therefore, these patients might be at high risk of foot ulcer development and might need much time to heal once foot ulcers occur. It is important to be proactive with foot risk evaluation and pay attention to nutritional status assessed with ODA in clinical practice, especially in older patients with a high-risk foot. Monitoring nutritional status in older patients with type 2 diabetes might be helpful to prevent future foot ulcers.

This study had several limitations. First, because of the study’s cross-sectional design, causal relationships could not be mentioned. Second, there is no information about the subjective nutritional indicators and sarcopenia assessed by skeletal muscle mass with body composition tests. Third, we categorized patients as low-risk and high-risk to perform multivariate analysis in this study. However, grouping patients with risk foot 1, 2, and 3 might lead to biased results, due to the heterogeneous characteristics and small sample size. Finally, this study was performed at a single institution, and all participants were Japanese. Therefore, whether our findings can be applied to other populations is uncertain.

## 5. Conclusions

In conclusion, it was shown that nutritional status assessed with ODA was significantly worse in patients with type 2 diabetes and high-risk foot in the older population.

## Figures and Tables

**Table 1 jcm-11-01314-t001:** Clinical characteristics of the participants.

	Group 0	Group 1	Group 2	Group 3	*p*
*n*	269	150	38	12	
Age (years)	64.2 ± 13.4	70.6 ± 11.1 *	73.6 ± 11.0 *	68.8 ± 12.4	<0.001
Male (%)	62.5	54.4	51.2	75.0	0.077
Duration of type 2 diabetes (year)	11.5 ± 10.0	16.6 ±11.4 *	22.7 ± 12.0 *^,†^	17.5 ± 8.9	<0.001
BMI (kg/m^2^)	25.0 ± 4.4	23.9 ± 4.8	24.4 ± 3.6	25.0 ± 5.0	0.225
Hemoglobin A1c (%)	7.2 ± 1.1	7.4 ± 1.2	7.7 ± 0.9	7.5 ± 1.0	0.142
Creatine (µmol/L)	69.7 ± 32.7	90.8 ± 86.9 *	91.9 ± 34.2 *	89.3 ± 36.4	0.002
Current smoking (%)	7.8	8.0	12.7	16.7	0.901
Statin use (%)	35.3	42.0	47.4	50.0	0.128
Hypertension (%)	54.6	70.7	81.6	83.3	0.001
Retinopathy (%)	22.3	40.0	42.1	41.7	<0.001
Nephropathy (%)	35.7	58.7	73.9	75.0	<0.001
Total cholesterol (mmol/L)	4.7 ± 0.9	4.5 ± 0.8	4.1 ± 0.5 *^,†^	4.1 ± 1.0	<0.001
Cholinesterase (U/L)	337.7 ± 95.6	309.4 ± 91.5 *	314.8 ± 92.9	311.2 ± 85.5	0.032
Serum albumin (g/dL)	4.2 ± 0.4	4.0 ± 0.4 *	3.9 ± 0.4 *	3.9 ± 0.6	<0.001
Hemoglobin (g/dL)	13.8 ± 1.6	12.8 ± 2.0 *	12.8 ± 1.8 *	13.8 ± 1.5	<0.001
Lymphocyte (count/mL)	2037 ± 857	1848 ± 714	1851 ± 848	1877 ± 589	0.134
CONUT	1 (1–3)	2 (1–5) *	3 (1–9) *	2 (1–4)	0.001

Continuous variables are presented as means ± 1 SD. Skewed variables are presented as medians (interquartile range). Categorical variables are presented as numbers (percentage). Group 0, no LOPS and no PAD; Group 1, LOPS or PAD; Group 2, LOPS + PAD or LOPS + foot deformity or PAD + foot deformity; and Group 3, LOPS or PAD and one or more of the following: history of a foot ulcer, a lower-extremity amputation (minor or major), or end-stage renal disease. LOPS, loss of protective sensation; PAD, peripheral artery disease; BMI, body mass index. * *p* < 0.05 vs. Group 0; and ^†^ *p* < 0.05 vs. Group 1.

**Table 2 jcm-11-01314-t002:** Comparisons of variables between low and high foot risk category in all patients.

	Low Foot Risk	High Foot Risk	*p*
*n*	269	200	
Age (years)	64.2 ± 13.4	71.2 ± 11.3	<0.001
Male (%)	62.5	55.0	0.147
Duration of type 2 diabetes (year)	11.5 ± 10.0	18.0 ±12.0	<0.001
BMI (kg/m^2^)	25.0 ± 4.4	24.1 ± 4.7	0.051
Hemoglobin A1c (%)	7.2 ± 1.1	7.4 ± 1.2	0.059
Creatine (µmol/L)	70.7 ± 35.3	88.4 ± 79.7	<0.001
Current smoking (%)	8.2	8.5	0.776
Statin use (%)	38.7	43.5	0.093
Hypertension (%)	57.2	73.5	<0.001
Retinopathy (%)	22.5	40.5	<0.001
Nephropathy (%)	37.9	62.0	<0.001
Total cholesterol (mmol/L)	4.7 ± 0.9	4.5 ± 0.8	0.005
Cholinesterase (U/L)	337.7 ± 95.6	313.1 ± 94.2	0.007
Serum albumin (g/dL)	4.2 ± 0.4	4.0 ± 0.4	<0.001
Hemoglobin (g/dL)	13.8 ± 1.6	12.9 ± 1.9	<0.001
Lymphocyte (count/mL)	2037 ± 857	1858 ± 724	0.017
CONUT	1 (1–3)	2 (1–4)	0.001

Continuous variables are presented as means ± 1 SD. Skewed variables are presented as medians (interquartile range). Categorical variables are presented as numbers (percentage). BMI, body mass index; CONUT, controlling nutritional status.

**Table 3 jcm-11-01314-t003:** Comparisons of variables between patients <75 and ≥75.

	<75 Years	≥75 Years	*p*
n	339	130	
Age (years)	61.1 ± 10.8	80.8 ± 4.3	<0.001
Male (%)	61.7	53.1	0.197
Duration of type 2 diabetes (year)	12.3 ± 10.5	20.4 ±11.8	<0.001
BMI (kg/m^2^)	25.2 ± 4.8	23.3 ± 3.8	<0.001
Hemoglobin A1c (%)	7.3 ± 1.2	7.4 ± 1.0	0.694
Creatine (µmol/L)	70.7 ± 53.3	88.3 ± 44.1	0.022
Current smoking (%)	9.1	6.2	0.191
Statin use (%)	38.6	47.7	0.101
Hypertension (%)	63.1	71.5	0.027
Retinopathy (%)	27.4	38.5	0.011
Nephropathy (%)	43.1	64.6	<0.001
Total cholesterol (mmol/L)	4.6 ± 0.9	4.5 ± 0.8	0.019
Cholinesterase (U/L)	339.3 ± 99.8	297.9 ± 82.8	<0.001
Serum albumin (g/dL)	4.1 ± 0.4	3.9 ± 0.5	<0.001
Hemoglobin (g/dL)	13.7 ± 1.6	12.6 ± 1.8	<0.001
Lymphocyte (count/mL)	2047 ± 833	1647 ± 584	<0.001
CONUT	1 (1–3)	2 (1–4)	<0.001

Continuous variables are presented as means ± 1 SD. Skewed variables are presented as medians (interquartile range). Categorical variables are presented as numbers (percentage). BMI, body mass index; CONUT, controlling nutritional status.

**Table 4 jcm-11-01314-t004:** (**a**) Comparisons of variables in patients ≥75 years. (**b**) Comparisons of variables in patients <75 years.

(a)					
	Group 0	Group 1	Group 2	Group 3	*p*
*n*	45	64	22	4	
Age (years)	79.6 ± 4.2	80.8 ± 4.3	83.8 ± 4.2 *	78.8 ± 3.8	0.021
Male (%)	57.8	51.6	45.5	75.0	0.510
Duration of type 2 diabetes (year)	16.0 ± 11.1	21.5 ±11.1	28.1 ± 13.3 *	17.8 ± 11.0	0.002
BMI (kg/m^2^)	22.8 ± 3.5	23.1 ± 3.6	23.9 ± 3.8	23.9 ± 2.9	0.576
Hemoglobin A1c (%)	7.3 ± 0.9	7.4 ± 0.9	7.9 ± 1.1	7.4 ± 1.5	0.058
Creatine (µmol/L)	70.6 ± 26.4	91.9 ± 55.1	93.6 ± 35.6	94.3 ± 56.8	0.153
Current smoking (%)	6.7	4.7	4.5	25.0	0.883
Statin use (%)	44.4	39.1	40.9	50.0	0.842
Hypertension (%)	60.0	65.6	90.9	75.0	0.048
Retinopathy (%)	17.8	42.2	45.5	50.0	0.003
Nephropathy (%)	48.9	57.8	81.8	75.0	0.012
Total cholesterol (mmol/L)	4.8 ± 1.0	4.4 ± 0.8	4.3 ± 0.6	4.4 ± 1.4	0.149
Cholinesterase (U/L)	305.5 ± 85.8	290.1 ± 84.4	307.6 ± 84.0	281.0 ± 92.3	0.779
Serum albumin (g/dL)	4.1 ± 0.6	3.9 ± 0.4	3.8 ± 0.3	3.5 ± 0.7 *	0.005
Hemoglobin (g/dL)	13.1 ± 1.6	11.9 ± 1.9 *	12.2 ± 1.8	13.1 ± 1.9	0.009
Lymphocyte (count/mL)	1692 ± 762	1613 ± 593	1702 ± 712	1873 ± 417	0.871
CONUT	2 (0–3)	2 (1–5)	3 (1–9)	5 (1–11)	0.049
**(b)**					
	**Group 0**	**Group 1**	**Group 2**	**Group 3**	* **p** *
*n*	224	86	16	8	
Age (years)	59.5 ± 11.5	64.1 ± 9.7 *	64.3 ± 7.0	62.6 ± 11.3	0.004
Male (%)	63.4	58.1	50.0	75.0	0.289
Duration of type 2 diabetes (year)	10.5 ± 9.5	12.9 ±10.4	17.6 ± 12.1 *	15.3 ± 9.7	0.007
BMI (kg/m^2^)	25.4 ± 4.5	24.5 ± 5.5	24.6 ± 3.9	25.6 ± 5.5	0.496
Hemoglobin A1c (%)	7.2 ± 1.2	7.4 ± 1.3	7.4 ± 0.7	7.5 ± 0.8	0.788
Creatine (µmol/L)	73.3 ± 38.1	76.3 ± 32.3 *	88.7 ± 41.5 *	81.8 ± 31.6	<0.001
Current smoking (%)	8.5	10.5	12.5	12.5	0.511
Statin use (%)	37.5	44.2	56.3	50.0	0.438
Hypertension (%)	56.7	74.4	68.8	87.5	0.029
Retinopathy (%)	23.2	38.3	37.5	37.5	0.075
Nephropathy (%)	35.7	59.3	56.3	75.0	<0.001
Total cholesterol (mmol/L)	4.7 ± 0.9	4.8 ± 0.9	4.2 ± 0.6	4.0 ± 0.9	0.011
Cholinesterase (U/L)	344.9 ± 96.2	323.7 ± 102.4	323.0 ± 109.6	331.1 ± 77.6	0.368
Serum albumin (g/dL)	4.2 ± 0.3	4.0 ± 0.5 *	3.9 ± 0.5 *	4.1 ± 0.3	0.002
Hemoglobin (g/dL)	13.9 ± 1.6	13.3 ± 1.9 *	13.4 ± 1.7	14.1 ± 1.2	0.038
Lymphocyte (count/mL)	2106 ± 860	2010± 758	1979 ± 980	1890 ± 650	0.729
CONUT	1 (1–3)	1 (1–3)	3 (1–7)	2 (1–3)	0.062

Continuous variables are presented as means ± 1 SD. Skewed variables are presented as medians (interquartile range). Categorical variables are presented as numbers (percentage). Group 0, no LOPS and no PAD; Group 1, LOPS or PAD; Group 2, LOPS + PAD or LOPS + foot deformity or PAD + foot deformity; and Group 3, LOPS or PAD and one or more of the following: history of a foot ulcer, a lower-extremity amputation (minor or major), or end-stage renal disease. LOPS, loss of protective sensation; PAD, peripheral artery disease; BMI, body mass index; CONUT, controlling nutritional status. *, *p* < 0.05 vs. Group 0.

**Table 5 jcm-11-01314-t005:** Multivariate-adjusted ORs (95% CI) for high-risk diabetic foot assessed with IWGDF.

	<75 Years		≥75 Years	
	OR (95%CI)	*p*	OR (95%CI)	*p*
Age	1.05 (1.02–1.08)	0.002	1.02 (0.93–1.12)	0.667
Male	1.82 (1.04–3.23)	0.037	2.68 (1.05–7.19)	0.038
Duration of type 2 diabetes	1.02 (0.99–1.04)	0.157	1.06 (1.01–1.10)	0.007
BMI	0.99 (0.93–1.05)	0.701	1.10 (0.96–1.25)	0.174
Hemoglobin A1c	1.32 (1.05–1.65)	0.022	1.53 (0.97–2.53)	0.157
Creatine	1.02 (1.01–1.03)	<0.001	1.01 (0.99–1.02)	0.319
Hypertension	1.52 (0.83–2.79)	0.223	1.78 (0.67–4.81)	0.241
Current smoking	1.53 (0.66–3.47)	0.301	2.24 (0.44–13.5)	0.332
CONUT	0.94 (0.87–1.01)	0.107	1.37 (1.05–1.86)	0.021

OR, odds ratio; CI, confidence interval; IWGDF, International Working Group on the Diabetic Foot; BMI, body mass index; CONUT, controlling nutritional status.

## Data Availability

The datasets used and/or analyzed during the current study are available from the corresponding author upon reasonable request.

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
