# Peer review of "Nutritional Status Assessed with Objective Data Assessment Correlates with a High-Risk Foot in Patients with Type 2 Diabetes"

_jcm, 2022, doi:10.3390/jcm11051314_

Round 1

Reviewer 1 Report

Thank you for your efforts developing the study.

Although it is a well-written manuscript, I believe it has a number of flaws that should be addressed before it is considered for publication:

You use the 2016 IWGDF guideline, please clarify in which time period you have recruited patients, as my recommendation is that you use the updated 2019 guidelines, which were published in 2020.

On the other hand, I recommend that you do a segregated analysis by risk level as it is not the same a patient with risk foot 1 than a patient with risk foot 3. The risk of suffering complications related to Diabetic Foot Syndrome is higher the higher the risk foot the patient presents. Grouping patients with risk foot 1, 2 and 3 may lead to biased results as they would be heterogeneous groups of patients.

Author Response

Dear Reviewers,

I give deep thanks for your kind and detailed review. Here, I submit revised manuscript and would like to answer for your comments. I hope that the revised manuscript meets your approval and will be more suitable for publication.

< Answer to reviewer 1>

Thank you for your efforts developing the study. Although it is a well-written manuscript, I believe it has a number of flaws that should be addressed before it is considered for publication:You use the 2016 IWGDF guideline, please clarify in which time period you have recruited patients, as my recommendation is that you use the updated 2019 guidelines, which were published in 2020.

Response

We give deep thanks for your important and kind review. As you pointed out, we have referred the report IWGDF guidelines updated 2019. We listed this report as new reference 8.

Reference

  1. Schaper NC, van Netten JJ, Apelqvist J, Bus SA, Hinchliffe RJ, Lipsky BA; IWGDF Editorial Board. (2020) Practical Guidelines on the prevention and management of diabetic foot disease (IWGDF 2019 update). Diabetes Metab Res Rev 36 Suppl 1:e3266. doi: 10.1002/dmrr.3266.

We reconsider and reanalyzed the data with this updated IWGDF guidelines. The number of the participants of Group 1 and Group 2 have been changed. We have changed Table 4-a and Table 4-b and have revised the description about the Results section. We have also revised the description about the section of Materials and Methods and section as below.

Materials and Methods

“Type 2 diabetes was diagnosed as previously reported [10], and diabetic foot risk was categorized into groups using the IWGDF classification as follows [8]: 0 (no loss of protective sensation (LOPS) and no peripheral artery disease (PAD)), 1 (LOPS or PAD), 2 (LOPS + PAD or LOPS + foot deformity or PAD + foot deformity), and 3 (LOPS or PAD and one or more of the following: history of a foot ulcer, a lower-extremity amputation (minor or major) and end-stage renal disease).”

Results

“Patients with group 1 and group 2 assessed using the IWGDF criteria were significantly older and had a longer duration of diabetes than those in group 0. Total cholesterol was worse in group 2 and cholinesterase was worse in group 1 significantly. Serum albumin level, hemoglobin and CONUT scores were significantly worse in group 1 and 2.”

“A stratified analysis between older (≥ 75 years) and younger groups showed that serum albumin was low in group 3 and hemoglobin was low in group 1 significantly in the older group (Table 4-a). In the group younger than 75 years of age, serum albumin levels in group 1 and 2 were low and hemoglobin was low in group 1 significantly (Table 4-b). In all age groups, there were no significant differences in BMI and HbA1c with or without foot risk. The proportions of hypertension and nephropathy had significant difference in each group, and the disease duration was significantly longer in group 2 in both the older and younger groups (Table 4-a, 4-b).”

On the other hand, I recommend that you do a segregated analysis by risk level as it is not the same a patient with risk foot 1 than a patient with risk foot 3. The risk of suffering complications related to Diabetic Foot Syndrome is higher the higher the risk foot the patient presents. Grouping patients with risk foot 1, 2 and 3 may lead to biased results as they would be heterogeneous groups of patients.

Response

Thank you for your suggestion. As you say, patients with a risk foot of 1 are not the same as patients with a risk foot of 3 and grouping patients by risk foot 1, 2, or 3 may lead to bias, so it is desirable to analyze separately for each risk level. Unfortunately, however, the number of the patients were not sufficient to perform statistical analysis. Thus, patients with risk foot 1, 2, and 3 were grouped together. According to your comment, we have added this point in the Discussion section described as below.

Discussion

“Third, we categorized patients as low-risk and high-risk to perform multivariate analysis in this study. However, grouping patients with risk foot 1, 2 and 3 might lead to biased results as their heterogeneous characteristics and small sample size.”

Yours sincerely,

Reviewer 2 Report

The manuscript provides a new point of preventing one of the main pathologies of world interest, diabetic foot.

As minor comments towards the authors, I expose:

  • The abstract does not include all the sections of the work, it hardly reflects the methodology carried out.

  • The introduction should include justification for why nutritional novels affect these patients.

  • The justification for the methodology is terse.

  • In the discussion, the authors should reflect on how the results provided by the work would improve the monitoring of these patients.

  • The conclusions are not a direct response to the objective.

Author Response

Dear Reviewers,

I give deep thanks for your kind and detailed review. Here, I submit revised manuscript and would like to answer for your comments. I hope that the revised manuscript meets your approval and will be more suitable for publication.

< Answer to reviewer 2>

The abstract does not include all the sections of the work, it hardly reflects the methodology carried out.

Response

We thank for your kind review. We revised the description about Abstract to reflect the Materials and Methods as below.

“Malnutrition and diabetes are likely to co-occur. There are few reports on the association between nutritional status and foot risk in patients with type 2 diabetes (T2D). Therefore, we aimed to investigate this relationship in this cross-sectional study. We investigated the relationships between Controlling Nutritional Status score, and foot risk, evaluated by the International Working Group on the Diabetic Foot (IWGDF), in consecutive patients with T2D. Patients were divided into groups 0 to 3 by IWGDF, and groups 1 to 3 were defined as high-risk groups. Among 469 patients, 42.6% (n=200) of them had high-risk foot. Patients with high-risk foot showed significantly older (71.2±11.3 vs. 64.2±13.4 years, p<0.001) and had a longer duration of diabetes (18.0 ±12.0 vs. 11.5±10.0 years, p<0.001) than those in the low-risk group. In the high-risk group, serum albumin level, total lymphocyte count, hemoglobin, and CONUT score were significantly worse, especially in older patients (≥75 years). Multivariate logistic regression analysis showed that there was a positive correlation between CONUT score and high-risk foot in older patients (OR, 1.37; 95% CI, 1.05-1.86; p=0.021). Our results indicated that nutritional status, assessed by ODA, correlated with high-risk foot, especially in older patients with T2D.”

The introduction should include justification for why nutritional novels affect these patients.

Response

We all thank your kind and precise review. We could add the description from the reference. Thank you.

“Malnutrition worsened underlying diseases and lead to unfavorable prognosis in older patients with diabetes [6]. Malnourished patients with diabetes have been shown to be twice as likely to have foot injuries compared with nourished patients [6].”

The justification for the methodology is terse.

Response

We give deep thanks for your kind review. We have added the description about the Materials and Methods precisely as below.

“The Controlling Nutritional Status (CONUT) score was calculated using the data of serum albumin levels, total cholesterol levels, and total lymphocyte counts [9]; albumin levels ≥3.5, 3.5> and >3.0, 2.99> and ≥2.5, and <2.5 g/dL were scored as 0, 2, 4, and 6 points, respectively; total lymphocyte count ≥1,600, 1599–1,200, 1,199–800, and <800/mm3 were scored as 0, 1, 2, and 3 points, respectively; and total cholesterol levels ≥180, 140–179, 100–139, and <100 mg/dL were scored as 0, 1, 2, and 3 points, respectively. The CONUT score was defined as the sum of scores, ranges from 0 to 12, with higher scores indicating a worse nutritional status.”

“DPN was diagnosed using the diagnostic criteria for diabetic neuropathy proposed by the Diagnostic Neuropathy Study Group [12]. Two or more abnormalities of three examination items were used to diagnose DPN: neuropathic symptoms such as neuropathic pain, paresthesia and numbness, decreased or absent ankle reflex (bilateral) and decreased distal sensation assessed by C128 Hz tuning fork without evident non-diabetic peripheral neuropathy. Diabetic retinopathy was diagnosed by an ophthalmologist as previously reported [13], and diabetic nephropathy was defined as nephropathy with urine microalbuminuria > 30 mg/gCre [14]. PAD was diagnosed if at least one of the following was confirmed: ankle brachial pressure index (ABI) < 0.9 or absence of two or more pedal pulses on palpation. Foot deformity and musculoskeletal abnormalities were examined to detect hallux valgus deformity, hammer/claw toes deformity and hallux limitus (limited motion at the metatarsophalangeal joint). “

In the discussion, the authors should reflect on how the results provided by the work would improve the monitoring of these patients.

Response

As you pointed out, we agreed your suggestion and have added the description in Discussion section as below.

“It is important to be proactive foot risk evaluation and pay attention to nutritional status assessed with ODA in clinical practice, especially in older patients with a high-risk foot. Monitoring nutritional status in older patients with type 2 diabetes might be helpful to prevent future foot ulcers.”

The conclusions are not a direct response to the objective.

Response

As your kind and important review, we have deleted the part of the description about the conclusion.

“In conclusion, it was shown that nutritional status assessed with ODA was significantly worse in patients with type 2 diabetes and high-risk foot in the older population.”

Yours sincerely,

Round 2

Reviewer 1 Report

Thank you for add the changes recommended. The manuscript has been improved.